# EMG Signs of Motor Units' Enlargement in Stroke Survivors

**Talita P. Pinto [1,2,3], Andrea Turolla [4,5], Marco Gazzoni [2,3], Michela Agostini [6] and Taian M. Vieira [2,3,*]**

1   Instituto D'Or de Pesquisa e Ensino (IDOR), Rio de Janeiro 22281-100, Brazil
2   PoliToBIOMed Lab, Politecnico di Torino, 10129 Turin, Italy
3   Laboratory for Engineering of the Neuromuscular System (LISiN), Department of Electronics and Telecommunications, Politecnico di Torino, 10129 Turin, Italy
4   Department of Biomedical and Neuromotor Sciences, Alma Mater University of Bologna, 40138 Bologna, Italy
5   Unit of Occupational Medicine, IRCCS Azienda Ospedaliero-Universitaria di Bologna, 40138 Bologna, Italy
6   Department of Neuroscience, Rehabilitation Unit, University—General Hospital of Padova, 35128 Padova, Italy
*   Correspondence: taian.martins@polito.it

**Abstract:** The degeneration of lower motoneurons has often been reported in stroke survivors, with possible collateral reinnervation from the surviving motoneurons to the denervated muscle fibers. Under this assumption, a stroke would be expected to increase the size of motor units in paretic muscles. We indirectly address this issue with electrical stimulation and surface electromyography, asking whether stroke leads to greater variations in the amplitude of M waves elicited in paretic muscles than in contralateral, non-paretic muscles. Current pulses at progressively greater intensities were applied to the musculocutaneous nerve, stimulating motoneurons supplying the biceps brachii of eight stroke patients. The size of increases in the amplitude of M waves elicited consecutively, hereafter defined as increments, was considered to evaluate changes in the innervation ratio of biceps brachii motor units following stroke. Our findings showed that patients presented significantly ($p$ = 0.016) greater increments in muscles of paretic than in non-paretic limbs. This result corroborates the notion that collateral reinnervation takes place after stroke, enlarging motor units' size and the magnitude of the muscle responses. Therefore, the non-invasive analysis proposed here may be useful for health professionals to assess disease progression by tracking for neuromuscular changes likely associated with clinical outcomes in stroke survivors, such as in the muscles' strength.

**Keywords:** stroke; motor unit; electromyography; M wave; muscle reinnervation

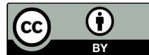

## 1. Introduction

The degeneration of motoneurons following stroke has been suggested to depend on a neurophysiological process called transneuronal degeneration [1,2]. According to this process, the interruption of the transmission of electrical impulses between neighboring neurons leads to the death of neurons [3]. The main evidence of such a degenerative process came from studies investigating the number of motor units in the muscles of stroke survivors through techniques often termed motor unit number estimation (MUNE) [1,2,4–6]. Findings from these studies indicate a significant decrease of about 20–60% in the number of muscles' motor units on the paretic side with respect to the non-paretic side. As a consequence, the loss of motor units may lead to muscle fiber reinnervation following a stroke.

There is evidence that reinnervation takes place after stroke through the collateral branching of the surviving motoneurons to the denervated muscle fibers [7–11]. In such a case, each surviving motoneuron may innervate a greater number of muscle fibers, possibly leading to an increased number of muscle fibers per motor unit (innervation ratio)

after stroke. Indeed, through an invasive analysis of single-fiber electromyography, studies [7,8,11] reported a significantly higher motor unit fiber density in hand muscles in the paretic limb of stroke survivors with respect to healthy muscles (either of the non-paretic side of stroke patients or of a control group). This increased fiber density presumably indicates that the paretic muscle may be characterized by a greater innervation ratio [12]. Besides the invasive analysis of single-fiber electromyography [7,8,11], such motor units' enlargement following stroke was also suggested from surface electromyography. For example, Kallenberg and Hermens [9] and Li et al. [10] observed larger EMG amplitudes for the biceps brachii muscle on paretic limbs compared to contralateral limbs during voluntary isometric contractions performed at relative force levels. Similarly, Vieira et al. [13] observed an increase in the size of the excited muscle region within the gastrocnemius muscle in paretic limbs. These authors suggest that the increased EMG response may be associated with muscle fiber reinnervation.

Assessing muscle reinnervation progress remains fundamental to comprehending factors that promote and hinder illness recovery. The muscle reinnervation process has been associated with the preservation of muscle strength [14]. On the other hand, the motor units' enlargement may impair the ability of a reinnervated muscle to produce force increments as finely as healthy muscles since a greater amount of muscle fibers innervated by motoneurons would be activated for similar synaptic inputs [15]. Therefore, the identification of techniques to track changes in the size of motor units (i.e., in the muscle's innervation ratio) would assist health professionals in assessing disease progression in stroke survivors. However, the methodologies applied in the above-mentioned studies investigating muscle reinnervation in stroke survivors present some limitations. Besides being an invasive technique, single-fiber electromyography assesses a relatively small portion of the entire pool of motor units in the muscle [16]. The other limitation is that the protocols considering voluntary contractions cannot be applied in hemiplegic patients (i.e., with complete paralysis of one side of the body), a common sequela of stroke [17].

Different methods involving incremental electrical stimulation, such as the "motor unit number estimation—MUNE" [18], the "electrophysiological muscle scan" [19], and the "motor unit size index—MUSIX" [14], have been proposed to indirectly assess structural adaptations on the neuromuscular system. The incremental stimulation technique gradually increases the stimulus intensity applied to a peripheral nerve to recruit successive motor units [18]. Since the motoneurons that innervate a muscle have different sizes and, therefore, different activation thresholds [20,21], it is possible to recruit them by gradually increasing the intensity of current pulses applied to the nerve [18]. As the stimulation intensity increases, more motoneurons are activated and, consequently, more muscle fibers are recruited. Such progressive recruitment can be observed through increases in the amplitude of massed action potentials (M wave) [18,19]. If no variations in the M-wave amplitude are observed as stimulation intensity increases, this indicates that a maximum M-wave response was elicited in the muscle, i.e., motor units corresponding to the pool of motoneurons stimulated were totally recruited [18].

Based on that, a potential method to non-invasively assess signs of muscle fiber reinnervation could be the combination of incremental neuromuscular electrical stimulation with high-density surface EMG. If the innervation ratio increases following stroke [7,8,11], it is reasonable to expect that similar increases in the intensity of nerve stimulation applied to both paretic and non-paretic muscles lead to greater variations in the M-wave amplitude for the paretic muscles. The advantage of investigating muscles' responses from electrically elicited contractions is the assurance of recruiting most, if not all, motor units in the evaluated muscles [22], in particular, because stroke survivors may be unable to voluntarily contract their paretic muscles to a greater extent. Additionally, the high-density surface EMG enables the sampling of action potentials from a wide muscle region, providing more representative myoelectric activity than conventional bipolar electrodes [23]. Therefore, applying these non-invasive techniques to stroke survivors could help clinicians, therapists, and researchers assess muscle reinnervation progress.

The current study aims at verifying whether increments in the amplitude of M waves elicited in muscles of stroke patients are greater in paretic than non-paretic limbs for similar, relative increases in stimulation intensity. Since the upper limbs are usually the most affected among stroke survivors [24], we address our question for the biceps brachii muscle.

## 2. Materials and Methods

### 2.1. Participants

Twenty stroke patients (14 men; range values; age: 42–84 years; body mass: 52–102 kg; height: 1.52–1.85 m) were recruited to participate in this study after providing written informed consent according to the Declaration of Helsinki. The experimental protocol was approved by the Ethical Committee of the S. Camillo Hospital (Lido di Venezia, Italy). Patients were classified according to the Oxford Community Stroke Project (OCSP) criteria, a clinical classification method that predicts the site of the infarct [25]. Then, they were screened and enrolled according to the following criteria: diagnosis of stroke (ischemic or hemorrhagic) confirmed through computed tomography or magnetic resonance imaging examinations by an experienced neurologist; the presence of motor impairment in the upper extremity due to stroke defined as a Fugl–Meyer Assessment for Upper Extremity [26] lower than 66 points; no associated traumatic brain injury; no history of orthopedic or neurological injury that might affect upper-extremity muscle function. Moreover, considering previous accounts suggesting motor unit reorganization occurs in the first months after stroke [7,8], patients with at least one month from stroke were included.

### 2.2. Motor Function Evaluation and Dominance Evaluation

The Fugl–Meyer Assessment for Upper Extremity [26] was used to assess the motor function of the biceps brachii muscles of the patients recruited in this study. We considered for analysis only the items of the scale related to the motor function of biceps brachii, within subsections I–IV from section A. Patients whose Fugl–Meyer evaluation was above 90% of the maximum score (i.e., >10 points) were excluded from the study (6 out of 20 patients), to ensure the biceps brachii muscles assessed in the paretic side were indeed functionally impaired.

Moreover, since our previous study [27] showed a significant dominance effect on muscles' responses from biceps brachii motor units of healthy young subjects, patients' dominance was also taken into consideration. Arm preference was evaluated for each patient through the *laterality quotient* of the Edinburgh Handedness Inventory [28], whereby patients were asked to answer the query according to the period before the stroke.

### 2.3. Experimental Protocol

After the Fugl–Meyer evaluation, 14 patients participated in the experimental protocol. They were comfortably seated with the upper limb under investigation secured to an isometric brace (Figure 1). For both paretic and non-paretic limbs, the forearm was held in a pronate position, the elbow joint was flexed at about 110° (180° being full extension), and the shoulder joint was abducted at about 45° (180° being full abduction). Visual inspection ensured the maintenance of the same joint angle for each patient's paretic and non-paretic sides.

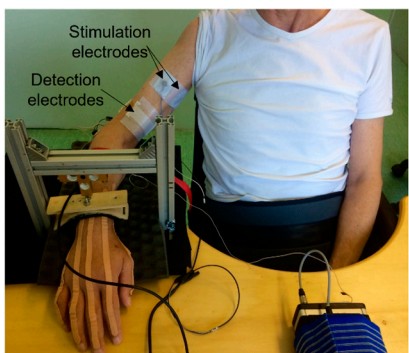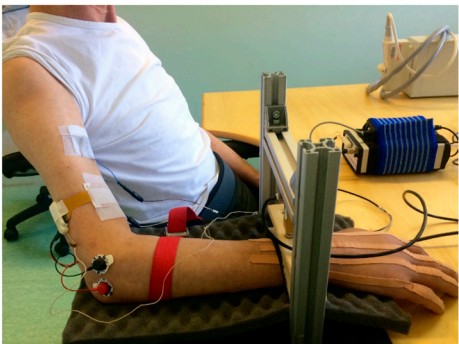

**Figure 1.** Frontal (**left image**) and lateral (**right image**) views of the experimental setup performed with the paretic limb of a patient. Electrodes for surface EMG detection and stimulation were positioned in the distal and proximal portions of the biceps brachii muscle, respectively. The tape attached to the patient's fingers is for rehabilitation purposes and, therefore, is not related to the experimental protocol.

The stimulation protocol consisted of a staircase stimulation profile, starting from the minimal current delivered by the stimulator (2 mA; Rehastim Science Mode, Hasomed, Magdeburg, Germany) until the maximal intensity tolerated by each patient. The current intensity was gradually increased automatically at the smallest possible current step (2 mA) allowed by the stimulation device. For each stimulation intensity, four biphasic, rectangular current pulses (100 μs per phase) were applied at 1 pps. The duration of the stimulation protocol depended on the maximal current intensity tolerated by each patient; it lasted 4.2 min at most. The experimental protocol was first conducted in the non-paretic limb and subsequently in the paretic limb, as all patients disclosed feeling more comfortable this way, and given our results, are not expected to depend on which side was assessed first. The total duration of the experiment was about one hour and a half.

### 2.4. Stimulation Electrode Positioning

A pair of stimulation electrodes (size 35 × 45 mm) was positioned proximally and arranged orthogonally to the muscle longitudinal axis (cf. Figure 1B in Pinto et al. [27]). Specifically, with the participant's arm secured to the isometric brace, the lateral and medial borders of biceps brachii, as well as the deltoid contour, were identified with ultrasound imaging (Echo Blaster 128, Telemed Ltd., Vilnius, Lithuania) and marked on the skin. Then, in relation to the biceps' longitudinal axis, the external edges of the cathode and anode electrodes were positioned just internally to the borders of the biceps' short and long heads, respectively. The superior edge of both electrodes was positioned distally to the deltoid inferior border.

### 2.5. EMG Recordings

A flexible printed circuit board (PCB) made up of a grid of 32 Ag/AgCl surface electrodes (4 × 8 arrangement; 3 mm diameter; 10 mm inter-electrode distance; LISiN-Politecnico di Torino, Turin, Italy) was used to sample multiple EMGs from both heads of the biceps brachii. Columns of electrodes were aligned parallel to the muscle's longitudinal axis. The grid of surface electrodes was positioned as distally as possible from stimulation electrodes, without covering the muscle tendon as identified with ultrasound imaging (Echo Blaster 128, Telemed Ltd., Vilnius, Lithuania). This technique was also applied to identify the junction between short and long biceps heads, where the midline between the fourth and fifth columns was centered. Surface EMGs were sampled in monopolar derivation (192 V/V gain; 10–750 Hz bandwidth amplifier; W-EMG LISiN-Politecnico di Torino, Turin, Italy) and digitized at 2441.4 Hz [29] with a 24-bit A/D converter. Before positioning detection electrodes, the patient's skin was cleaned with an abrasive paste (Nuprep, Weaver and Company, Aurora, CO, USA). An external trigger signal issued

with the stimulation pulses was recorded as an auxiliary signal with the EMG amplifier and used to identify M waves.

### 2.6. Assessment of Muscle Responses in Paretic and Non-Paretic Limbs

Firstly, monopolar EMGs were visually inspected to identify channels with contact problems or powerline interference. Low-quality signals, if present, were replaced with the linear interpolation of the neighbor channels [30]. Monopolar EMGs signals were then band-pass filtered with a fourth-order Butterworth filter (10–400 Hz cutoff; bidirectional filter). Single-differential EMGs were obtained by differentiating monopolar signals along consecutive rows. M waves were triggered from EMGs (30 ms epochs) and then averaged across the four stimulation pulses, separately for each channel and stimulation intensity. The presence of innervation zones, either in the first or between the first and second rows of the grid, was observed for 9 out of 28 muscles assessed. Thus, given that the innervation zone leads to a spurious decrease in the amplitude of surface EMGs [23], channels over the innervation zone were excluded, and only M waves with the greatest peak-to-peak amplitude in each column of the grid were considered for analysis. Amplitude values were then averaged across columns, producing a single, representative biceps brachii response per stimulation intensity.

Variations in the mean peak-to-peak amplitude with stimulation intensity were considered to assess whether biceps brachii responses are greater on the paretic than the non-paretic side. First, the range of current intensities leading to the smallest and greatest M waves was identified from the distribution of peak-to-peak amplitudes obtained for all stimulation intensities. The highest current intensity below which the peak-to-peak amplitude was smaller or equal to the first amplitude mode was then identified and defined as the *motor threshold*. Likewise, the smallest current intensity over which peak-to-peak amplitude equaled or exceeded the second or last amplitude mode was identified and defined as the *maximal muscle response*. These values define the current range (cf. shaded rectangles in Figure 2A,C) within which increases in stimulation intensity resulted in increased muscle response; i.e., increased M-wave amplitude. Because the current range differed between patient limbs, the increases of 2 mA steps in stimulation intensity represented a different percentage of the current range for paretic and non-paretic muscles. Therefore, to ensure like-with-like comparisons, the amplitude of M waves obtained from the limb with the smaller current range was linearly interpolated considering a fixed, stimulation step smaller than 2 mA (Figure 2C). This new stimulation step was computed as the value of the smaller current range divided by the number of stimulation levels within the greater current range between limbs. Then, the differences between peak-to-peak values (*increments*) obtained for consecutive stimulation levels were computed and normalized with respect to the difference between the amplitudes corresponding to the *maximal muscle response* and the *motor threshold*. *Increments* were then averaged across all current levels (hereafter referred to as *increment amplitude*) and considered to assess limb differences in biceps brachii responses.

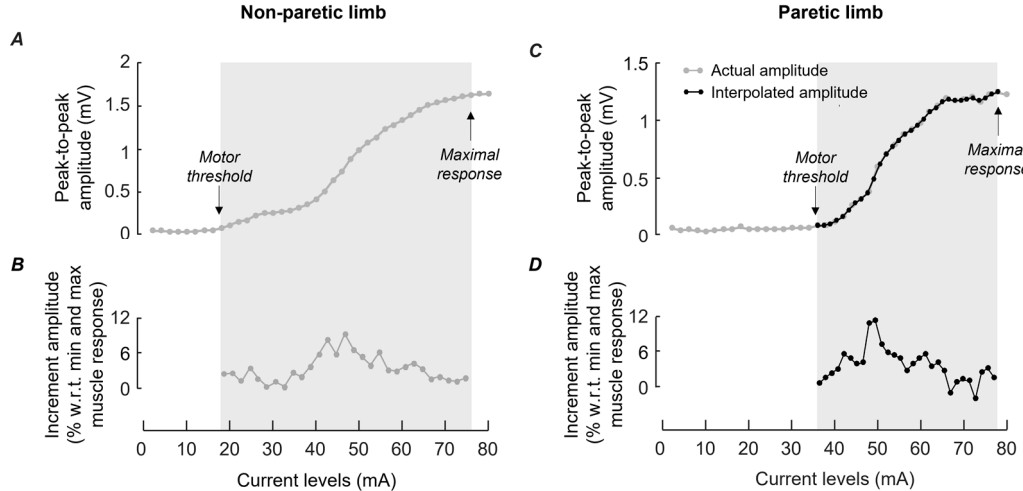

**Figure 2.** M-wave peak-to-peak amplitude for each current intensity is shown for the non-paretic (**A**) and paretic (**C**) limbs of a single, representative patient. Gray circles correspond to amplitude values averaged across the four stimulation pulses and channels. Black circles correspond to the interpolated amplitude values within the paretic limb's current range (shaded rectangle). After linear interpolation, the number of increments within the current range of the paretic limb increases from 21 to 29, to match the number of stimulation levels of the non-paretic limb. (**B,D**) show the normalized amplitude of increments between consecutive stimulation levels for the non-paretic and paretic limbs, respectively.

### 2.7. Statistics

Data from 6 out of 14 patients were discarded from the analysis because the M-wave amplitude increased indefinitely with stimulation intensity, producing a unimodal distribution of peak-to-peak values. The remaining eight patients were then clustered into two groups, according to different patterns observed from paretic and non-paretic muscles, i.e., according to whether the *increment amplitude* values were greater (*n* = 6) or smaller (*n* = 2) in the paretic than in the non-paretic muscle. Clusterization is a common approach in the literature investigating structural alterations in the neuromuscular system following diseases such as stroke and amyotrophic lateral sclerosis [10,14]. It allows us to attenuate the effects of a potential confounding factor in the study's outcome—the lack of muscle fiber reinnervation after the disease. Since the hypothesis of the present study considers that muscle fiber reinnervation takes place following stroke, to avoid the above-mentioned confounded factor, we excluded the two patients of the second group from the statistical analysis. Moreover, considering our small sample size (*n* = 6) and the inter-subject difference related to dominance and paretic limb (i.e., four patients had the paretic limb as the dominant one, while two patients had the paretic limb as the non-dominant one), it was not possible to group patients according to dominance for statistical analysis.

After verifying that the data distribution was Gaussian (Shapiro–Wilk's W test, *p* < 0.05) for all parameters evaluated, the parametric statistic was used to test our hypothesis. Student's *t*-test for dependent samples was applied to evaluate differences in *increment amplitudes* and current ranges between paretic and non-paretic muscles. The level of statistical significance was set at 5%, and data were reported using parametric descriptors.

### 3. Results

The demographic and clinical data of the patients evaluated are presented in Table 1. The six patients analyzed had the right limb as the dominant one, according to the median laterality quotient score (mean ± standard deviation: 83.7 ± 21.9%). The mean stimulation intensities across subjects, corresponding to the *motor threshold* and *maximal muscle*

*response* for biceps brachii were, respectively, 21.7 ± 6.4 mA and 79.3 ± 21.3 mA for the non-paretic limb, and 30.7 ± 15.9 mA and 86.3 ± 19.7 mA for the paretic limb.

**Table 1.** Demographic, clinical data, and increment amplitude values of the evaluated stroke survivors.

| ID | Sex | Age (yo) | Stroke Type | Paretic Limb | LQ Score (%) | Time since Stroke (Months) | FMA-UE Score | Inc Amp Non-Paretic Limb (%) | Inc Amp Paretic Limb (%) |
|---|---|---|---|---|---|---|---|---|---|
| 1 | M | 48 | hemorrhagic | right | 81.0 | 19 | 5/12 | 3.5 | 4.8 |
| 2 | F | 68 | ischemic | right | 90.5 | 10 | 4/12 | 4.0 | 4.1 |
| 3 | M | 61 | hemorrhagic | left | 90.5 | 39 | 10/12 | 1.4 | 1.6 |
| 4 | F | 71 | ischemic | right | 41.7 | 1 | 3/12 | 3.0 | 3.7 |
| 5 | M | 59 | ischemic | right | 100.0 | 2 | 2/12 | 2.3 | 3.1 |
| 6 | M | 70 | ischemic | left | 100.0 | 43 | 6/12 | 2.0 | 3.2 |

M: male. F: female. yo: years old. FMA-UE: Fugl–Meyer Assessment for Upper Extremity considered only items related to biceps brachii function. LQ: laterality quotient. Inc amp: increment amplitude.

*Differences between Muscle Responses in Paretic and Non-Paretic Limbs*

When comparing *increment amplitudes* between paretic (3.4 ± 1.1%) and non-paretic (2.7 ± 1.0%) limbs, the Student's *t*-test revealed greater values in paretic muscles ($p$= 0.016; $n$ = 12; 6 subjects × 2 limbs; Figure 3A). More specifically, *increment amplitude* values differed between limbs by up to 1.3%. The greater increment amplitudes indicate more muscle fibers were activated in paretic than non-paretic limbs for similar, relative increases in the stimulation intensity. This finding suggests that the analysis proposed in this study detected limb differences in the organization of the biceps brachii motor units, likely associated with a muscle reinnervation process due to stroke. Additionally, the statistical analysis also revealed no significant difference in the current range between sides ($p$ = 0.72; $n$ = 12; 6 subjects × 2 limbs; Figure 3B). Half of the patients showed greater ranges in the non-paretic than paretic limb, differing from 4 to 16 mA (i.e., from two to eight stimulation levels), and for the other half the current range was 2 to 18 mA (i.e., one to nine stimulation levels) greater in the paretic limb.

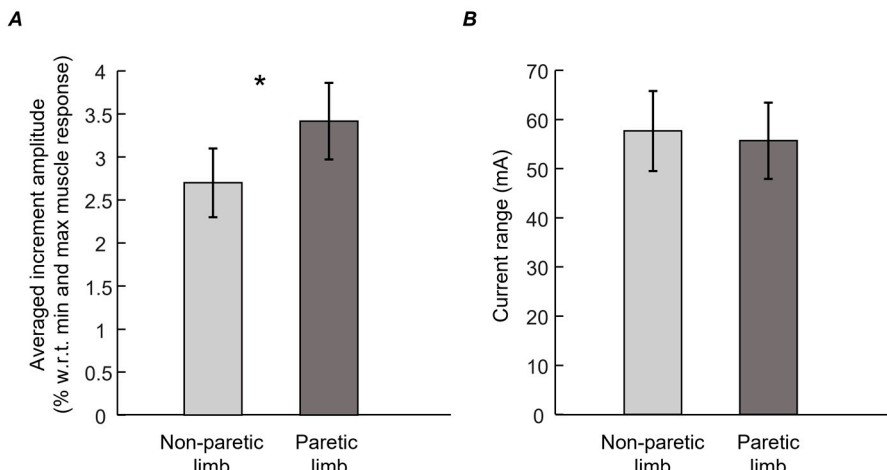

**Figure 3.** Boxplots in panels (**A**,**B**) show the *increment amplitudes* and current ranges calculated for non-paretic and paretic limbs of patients, respectively. The asterisk indicates a statistically significant difference ($p$ < 0.05) between limbs.

## 4. Discussion

In this study, we investigated whether, during incremental stimulation, changes in the M-wave amplitude between consecutive stimulation levels are greater in the biceps brachii muscle in paretic than non-paretic limbs of stroke survivors. M waves were assessed with high-density surface EMGs detected distally from the biceps brachii. The current intensity was gradually increased up to the maximal intensity tolerated by each patient. Our findings showed significantly greater changes in the amplitude of M waves between consecutive stimulation intensities in the paretic than non-paretic limbs of the evaluated patients.

### 4.1. Preliminary Considerations

To ensure there was an effect of stroke on the biceps brachii muscles evaluated in this study, we decided to disregard patients scoring greater than 90% in the Fugl–Meyer maximum score (i.e., patients with very mild or no motor impairment). In addition, methodological issues were considered to guarantee representative and selective EMGs of most biceps brachii motor units: (i) stimulation electrodes were positioned to ensure a similar charge density [31] impinged upon the nerve trunk and the nerve primary motor branches [32,33]; (ii) differential EMGs were sampled with a relatively small inter-electrode distance, attenuating any contribution from e.g., brachialis [34,35]; (iii) the linear interpolation in the stimulus–response curve allowed to compensate for the inability to increase the stimulation intensity by the same, relative amount between limbs. This is a conservative procedure given that across participants we observed a consistently smaller current range for the paretic limb before the interpolation.

### 4.2. Variations in M-Wave Amplitude Differed between Biceps Brachii of Paretic and Non-Paretic Limbs

Structural adaptations in the motor units' size of muscles affected after stroke were indirectly investigated by comparing the size of increments in the amplitude of M waves elicited in the biceps brachii of paretic and non-paretic limbs of stroke survivors. The M wave is the summation of action potentials of the synchronously activated muscle fibers in a muscle which is recorded from the surface electromyography [36]. Variations in the M-wave amplitude are usually assumed to reflect a change in the number of motor units recruited [18] and, therefore, in the number of muscle fibers activated. If the motor units' size (i.e., the innervation ratio) increases due to a reinnervation process, greater variations in the M-wave amplitude would be expected when motor units of a reinnervated muscle are gradually recruited during an incremental stimulation protocol [18,19]. Thus, assuming collateral reinnervation takes place in muscles affected after stroke [7,8,11], we expected that increases in M-wave amplitude, evaluated by means of median increments (i.e., *increment amplitude)*, would be greater in paretic rather than non-paretic muscles for similar, relative increases in current intensity. Our findings indicated that patients presented significantly greater *increment amplitudes* in muscles of paretic with respect to non-paretic limbs ($p = 0.016$) (cf. Figure 3A). The average difference between *increment amplitudes* of limbs was about 0.7%. Thus, increases in M-wave amplitude seemed to have been progressively larger in paretic than non-paretic biceps brachii for similar, relative increases in the current intensity.

Side differences in *increment amplitudes* may have been even greater for five out of the six patients analyzed. Two factors probably bias patients' results, upper-limb dominance and the interpolation of stimulus–response curves. In our previous study [27], we performed the same experimental protocol with 16 healthy, young subjects and found significantly smaller *increment amplitude* in the biceps brachii of dominant than non-dominant limbs. Interestingly, for four patients evaluated in the present study the dominant limb was the paretic one and, therefore, increment amplitudes were greater in the dominant than the non-dominant limb. These results were contrary to those found for healthy

subjects in Pinto et al. [27], and, therefore, patients' dominance probably underestimated side differences in *increment amplitudes*. In addition, for two patients (one within the just mentioned group), the current range was 16 mA greater in non-paretic than paretic limbs. It means linear interpolation was applied in the stimulus–response curve of paretic muscles, probably underestimating their *increment amplitudes*, as mentioned above. These findings corroborated the notion that collateral reinnervation occurs after a stroke, increasing motor units' size and, therefore, the magnitude of the muscle responses in the paretic limb. Previous accounts [7,8,11] indeed showed higher fiber density in muscles affected after stroke with respect to healthy, non-paretic muscles.

As above-mentioned in the statistics section, two stroke survivors were excluded from the analysis because *increment amplitudes* were smaller in paretic with respect to non-paretic muscles. These results indicate a muscle reinnervation process may not have occurred in the paretic biceps brachii of these patients. Such inter-subject variability observed in our findings regarding side differences of neuromuscular responses in stroke survivors was also reported by previous studies. Briefly, Kallenberg and Hermens [9] and Li et al. [10] analyzed the absolute amplitude values of surface EMGs recorded in muscles of paretic and non-paretic limbs of stroke patients during isometric contractions performed at different force levels. On the one hand, some patients they evaluated showed greater muscle responses in paretic than non-paretic limbs, suggesting a collateral reinnervation process. On the other hand, the other part of the patients presented opposite results, i.e., muscle responses were smaller on the paretic than non-paretic sides. Kallenberg and Hermens [9] suggested that the greater responses observed in non-paretic muscles may be explained by increased fiber diameter due to overuse. In contrast, Li et al. [10] considered muscle fiber atrophy a possible reason for the smaller responses observed in paretic muscles. In the present study, however, differences between the muscle fibers' diameters in paretic and non-paretic biceps brachii may not explain our findings since M-wave amplitudes were normalized with respect to the minimal and maximal muscle responses, compensating the effect of anatomical differences between limbs on the surface EMGs. Therefore, increment amplitudes values were not affected by possible muscle fiber atrophy after stroke.

Finally, by observing the individual patient results in Table 1, increment amplitude values for paretic limbs seem not to be associated with the time elapsed since the stroke onset and indicate that muscle reinnervation may have already occurred after a few months following the stroke. This assumption corroborates with previous accounts that observed increased fiber density in hand muscles affected after a few months that patients had a stroke [7,8].

### 4.3. Physiological Implications

Notwithstanding the reduced sample of patients evaluated in this study, we verified greater variations in the amplitude of M waves elicited in paretic than non-paretic biceps brachii. Such findings might provide evidence of changes in the organization of the neuromuscular system following stroke, specifically of increases in the innervation ratio of paretic muscles, probably due to muscle fiber reinnervation. According to a recent longitudinal study investigating motor unit size index (MUSIX) in patients with amyotrophic lateral sclerosis (ALS), the muscle reinnervation process is associated with the clinical preservation of muscle strength [14]. Motor units' enlargements, however, likely account for stroke survivors' inability to finely control movements with the paretic limb. The force generated by single motor units with respect to total muscle force reflects the quantal increment in muscle force, indicating how finely force may be regulated. In addition, the amount of force produced by individual units is proportional to the number of muscle fibers they supply [15], i.e., the smaller the number of muscle fibers innervated by a single motoneuron, the smaller the force produced per motor unit. Therefore, a muscle with a relatively low innervation ratio would produce finer increments in force than muscles with higher innervation ratios for similar synaptic inputs. Our group results showed that

the side difference in *increment amplitudes* was about 0.7% (cf. Figure 3A). Such a value can indicate the greater amount of additional muscle fibers elicited in the paretic muscle with respect to the non-paretic muscle for similar, relative increases in the stimulation intensity. In this hypothesis, if we translate this side difference in terms of incremental force, the biceps brachii of paretic limbs would produce greater relative force increments with respect to non-paretic limbs. Hence, the motor units' enlargement following a stroke may preserve the muscle strength but also hinder the ability of stroke survivors to produce increments in force as finely as healthy muscles.

The analysis proposed in the present study, combining surface EMG with incremental electrical stimulation, seems to provide an effective means for assessing structural adaptations in the neuromuscular system of stroke survivors. Previous accounts investigated muscle reinnervation in stroke survivors by applying protocols that may not be suitable in the clinical field. The use of invasive techniques (e.g., single-fiber electromyography) poses greater risk and discomfort for patients than non-invasive methods, besides probably hindering their recruitment. Another limitation of the methodologies previously applied is to perform assessments requiring voluntary contractions [9,10]. A common sequela of stroke is a complete or partial paralysis of one side of the body (hemiplegia and hemiparesis, respectively) [17]. Therefore, protocols involving neuromuscular electrical stimulation allow the evaluation of stroke survivors unable to contract the muscles voluntarily.

Hence, the experimental protocol applied in the present study is expected to further our understanding of stroke-induced changes as well as to help clinicians and therapists monitor disease progression and rehabilitative interventions in stroke survivors. Our findings indeed suggested changes in the organization of the neuromuscular system in the patients evaluated.

### 4.4. Study Limitation

Given that previous accounts suggest after stroke the number of motor units may be, on average, 20–60% smaller in paretic than non-paretic muscles [1,2,4–6], one could expect a clear difference in the M-wave responses between paretic and non-paretic biceps brachii evaluated in this study. The subtle side difference we observed in the *increment amplitude* (Figure 3A) may cause doubt as to whether the stimulus–response curve analysis effectively assessed structural adaptations of muscles affected after a stroke. However, before drawing such a conclusion, an important consideration must be highlighted: we do not know the degree of collateral reinnervation in the paretic muscles of the patients we evaluated. Suppose the number of motor units in a muscle is reduced by 50% after a stroke. In that case, this does not imply that survivor motoneurons would reinnervate all muscle fibers denervated in this muscle. Perhaps only part or none of the muscle fibers denervated are reinnervated. Therefore, more investigations in a greater sample of stroke survivors with similar characteristics (e.g., age and time since stroke onset) are necessary to confirm the effectiveness of the stimulus–response curve analysis performed in the present study.

**Author Contributions:** Conceptualization, T.P.P. and T.M.V.; methodology, T.P.P. and T.M.V.; validation, T.P.P. and T.M.V.; formal analysis, T.P.P.; investigation, T.P.P., A.T. and M.A.; resources, M.G., A.T. and M.A.; data curation, T.P.P.; writing—original draft preparation, T.P.P. and T.M.V.; writing—review and editing, A.T., M.G. and M.A.; visualization, T.P.P. and T.M.V.; supervision, M.G. and T.M.V.; project administration, T.P.P., A.T. and M.A.; funding acquisition, T.P.P., T.M.V. and M.G. All authors have read and agreed to the published version of the manuscript.

**Funding:** This research was funded by Compagnia di San Paolo and Fondazione C.R.T. T.P.P. was the recipient of a scholarship provided by Coordenação de Aperfeiçoamento de Pessoal de Nível Superior/Ciência semFronteiras/Processo n◦ BEX 9130/13-3. The funders had no role in study design, data collection, or analysis; the decision to publish; or manuscript preparation.

**Institutional Review Board Statement:** The study was conducted in accordance with the Declaration of Helsinki, and approved by the Ethics Committee of IRCCS San Camillo (protocol code 2016.21—M_Waves—20/12/2016).

**Informed Consent Statement:** Informed consent was obtained from all subjects involved in the study.

**Data Availability Statement:** The datasets used and/or analyzed during the current study are available from the corresponding authors upon reasonable request.

**Acknowledgments:** We thank Giorgia Pregnolato and Francesca Baldan for their support during data collection.

**Conflicts of Interest:** The authors declare no conflict of interest.

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
