# Peer review of "EMG Signs of Motor Units’ Enlargement in Stroke Survivors"

_applsci, doi:10.3390/app13042680_

Round 1

Reviewer 1 Report

Review

EMG signs of motor units’ enlargement in stroke survivors

Dear authors,

It was clear to understand your scientific contribution along the paper. I’ve talked about the main subject of the paper with my PhD students (physiotherapists) and it was a novelty for them. This study will help a large community of health professionals and better life quality for the subjects involved.

L. 37 – If you have a reason for each of the four references then keep them, otherwise use only the most relevant ones.

L. 44 and 65 – Just confirm the apostrophe in “muscle”.

L. 84 – Please, avoid inserting results and conclusion here in the introductory text. But they are very well presented in L. 266-70.

L. 91 – It was 20, but you informed only 14. Only in L. 119 you presented exclusion criteria by Fugl-Meyer evaluation. Please, fix the information in line 91.

L. 119 – The position is well described.

L. 128 – Figure 1 – What are the bands attached to the fingers? In the beginning, I thought they were a kind of array of electrodes.

L. 128 and 257 – Considering the "increments" is the difference between peak-to-peak, the unit is mV. If you agree with this, the "increment %" and "increment amplitude" have the same unit, I think it is important to nominate it in both y-axes in graphs 2B and 3A.

L. 263 – Insert “biceps brachii muscle” to rescue the idea presented in the introduction (L. 58).

L. 277 – 85 – It looks like a methods repetition without bringing novelty to the reader. Consider summarizing or cutting off it.

L. 356-8 – Just to say that is very well explained.

Reviewer 2 Report

1. The abstract part of the manuscript needs to highlight the findings and contributions of the paper, and the word "effective indirect evaluation" is inappropriate. Please specify the structural adaptability of nervous system.

2. Please explain the rationality of the experimental method or compare it with other experimental methods. It is suggested to briefly describe the necessity of the experimental steps, make a clear and concise analysis of the experimental techniques, and link them with the objectives of the paper.

3. In the results of the paper, please specify what the experimental data illustrate, what kind of results are obtained from the experimental data, and where the contribution to the paper is embodied.

4. Discussion and conclusion. It is suggested to add some specific descriptions, which can not only explain the electromyogram performance of imitating the increase of stroke survivors' motor units, but also explain the necessity of research.

In conclusion,  this paper can be accepted after answering the questions proposed above.

Reviewer 3 Report

Pinto et al. studied the M amplitude of paretic biceps brachii of stroke patients using multi-channel sEMG. Probably due to the deficiency of the trophic factor provided by the upper motor neurons, secondary neuronal loss in the anterior horn will occur in stroke patients. Under such an assumption, the authors compared the M amplitudes of the paretic and non-paretic limbs and found a relatively small but statistically significant incremental increase in the M amplitude in the paretic limb. Specifically, they showed more significant amplitude variations that corroborate the notion that collateral reinnervation will occur after a stroke. They planned their experiment carefully, and their methodology seemed sound. Their finding is interesting since it suggests the possibility that impairment of upper motor neurons due to stroke is accompanied by the loss of lower motor neurons. 

However, the reviewer concerns that the authors excluded two patients with incompatible findings. This fact might limit the external validity of their results.

Collateral reinnervation is one of the critical events found in ALS patients. In general, reinnervation will occur in the chronic denervation stage. It will be helpful if each % of increment is shown in Table 1 so that the readers can compare the duration of the disease and the degree of variation of M amplitude. As the authors mentioned, the atrophic muscle fiber elicits only a small amplitude in the advanced stage. Therefore, it is likely that the longer the disease duration, the more significant the variation. 

The reviewer cannot understand what the sign of “÷” such as “42÷84 years” in line 91, stands for. A simple misspelling?

Round 2

Reviewer 2 Report

Lines 66-69: I recommend that you follow each example with the corresponding reference

Lines 124-128: I suggest that you should describe the section in more detail. You should explain more clearly to the reader, your research contribution.

Line 191: Please add the material, type, manufacturer, and city of the surface EMG electrodes used.
